**Subject Category:**
Biology (whole organism)

evolution/developmental biology/molecular biology

opsin, Midas cichlid, amphilophus, Nicaragua, fluorescent *in situ* hybridization, visual system

**Author for correspondence:**
Julián Torres-Dowdall
e-mail: julian.torres-dowdall@uni-konstanz.de

# Reverting ontogeny: rapid phenotypic plasticity of colour vision in cichlid fish

Andreas Härer[1], Nidal Karagic[1], Axel Meyer[1] and Julián Torres-Dowdall[1,2]

[1]Zoology and Evolutionary Biology, Department of Biology, and [2]Zukunftskolleg, University of Konstanz, Konstanz, Germany

 AH, 0000-0003-2894-5041; NK, 0000-0003-3575-3558; AM, 0000-0002-0888-8193; JT-D, 0000-0003-2729-6246

Phenotypic plasticity, particularly during development, allows organisms to rapidly adjust to different environmental conditions. Yet, it is often unclear whether the extent and direction of plastic changes are restricted by an individual's ontogeny. Many species of cichlid fishes go through ontogenetic changes in visual sensitivity, from short to long wavelengths, by switching expression of cone opsin genes crucial for colour vision. During this progression, individuals often exhibit phenotypic plasticity to the ambient light conditions. However, it is commonly assumed that once an adult visual phenotype is reached, reverting to an earlier ontogenetic state with higher sensitivity at shorter wavelengths is not common. In this study, we experimentally demonstrate that four-month-old Midas cichlid fish (*Amphilophus astorquii*) show plasticity in single cone opsin expression after experiencing drastic changes in light conditions. Resulting shifts of visual sensitivity occurred presumably in an adaptive direction—towards shorter or longer wavelengths when exposed to short- or long-wavelength light, respectively. Single cone opsin expression changed within only a few days and went through a transitional phase of co-expression. When the environment was experimentally enriched in long-wavelength light, the corresponding change occurred gradually along a dorsoventral gradient within the retina. This plasticity allowed individuals to revert earlier ontogenetic changes and return to a more juvenile visual phenotype demonstrating previously unrecognized insights into temporal and spatial dynamics of phenotypic plasticity of the visual system in response to ambient light.

## 1. Introduction

Phenotypic plasticity, the ability to respond to environmental variation [1], is ubiquitous in nature but varies substantially among species. Particularly, adaptive phenotypic plasticity is an

important mechanism to approximate local fitness optima [2,3], also in adjusting the visual system to varying light conditions [4–6]. In vertebrate retinas, light absorption is achieved by visual pigments consisting of an opsin protein and a chromophore [7]. A visual pigment's wavelength of maximum sensitivity is determined by the type of chromophore and the structure of the opsin protein [8]. Cichlid fishes are an emerging model system in visual ecology since they have, as several other teleosts [9], a rich set of seven cone opsins (responsible for colour vision) compared with humans who only have three. In particular, cichlids have two types of cone cells, single and double cones, which express short (sws1, sws2b, sws2a) and medium- to long-wavelength-sensitive opsins (rh2b, rh2aβ, rh2aα, lws), respectively [10]. Moreover, opsin expression patterns are highly variable among species, suggesting that differences in colour vision evolve rapidly [10,11]. Accordingly, adaptive evolution and diversification of cichlids' visual systems occur mostly by expressing different subsets of cone opsins [11–13], which might have been facilitated by phenotypic plasticity of opsin expression patterns.

During ontogeny, many cichlid species shift opsin expression from short- to long-wavelength sensitivity [10]. Phenotypic plasticity in opsin expression seems to be rather common during this ontogenetic progression but appears to be restricted once the adult visual phenotype has been reached [5,14], but see [6,15]. Amphilophus astorquii, our study species, shows a paedomorphic visual phenotype, which might allow this species to maintain some degree of plasticity into adulthood, at least in the same direction as the ontogenetic progression (towards longer wavelengths) [4]. Yet, it remains unclear to what extent phenotypic plasticity of opsin expression can override typical ontogenetic changes in the long term and whether cone opsin expression patterns can also be reverted to a previous ontogenetic state.

Hence, A. astorquii provides an excellent system to investigate whether ontogenetic change constrains extent or direction of phenotypic plasticity of opsin expression. In this study, we found that (i) phenotypic plasticity occurs rapidly within a few days, (ii) plasticity can appear in the same but also the opposite direction of the ontogenetic progression in opsin expression and (iii) plastic changes seem to be produced by spatial modifications of opsin expression along the retina's dorsoventral axis.

## 2. Material and Methods

Second-generation laboratory-bred descendants of wild-caught A. astorquii were reared under light at the short (blue, 450 nm) or long (red, 630 nm) extremes of the visible light spectrum [4]. After hatching, siblings were randomly assigned to one of the two light treatments. At an age of four months—118 days post-hatching (dph)—fish in each treatment were randomly divided into two groups and were either retained in the same treatment or were exchanged between the treatments (figure 1a). From this point onwards for 14 days, one individual from the exchange groups (blue-to-red, red-to-blue) was collected every day to monitor changes in cone opsin expression. At the beginning and the end of the experimental period, retinas were collected from some individuals of the blue and red treatments ($n = 6/$ light treatment). Fish were sacrificed with an overdose of MS-222 (400 mg l$^{-1}$) and subsequent cervical dislocation. One retina per individual was stored in RNAlater for qPCR and one was fixed (4% paraformaldehyde in PBS) for fluorescence in situ hybridization (FISH). Total RNA was extracted (RNeasy Mini Kit; Qiagen, Hilden, Germany) and 500 ng were reverse transcribed (GoScript™ Reverse Transcription System; Promega, Madison, Wisconsin, USA). Expression levels of opsin genes were based on qPCR data of two technical replicates and mean threshold cycle (Ct) values were used for subsequent analysis. Before and after the 14-day experimental period, expression of cone opsins was calculated proportionally to overall cone opsin expression (incorporating qPCR efficiencies) as previously described [4,14]. For fish reared under either blue or red light for 118 and 132 days, we tested whether treatment or age has an effect on cone opsin expression by using a nonparametric equivalent to a two-way ANOVA, the Scheirer–Ray–Hare test [17]. All linear regression models were tested for heteroscedasticity with Breusch–Pagan tests [18]. During the 14-day experimental period, proportional expression was separately calculated for the predominantly expressed opsins in single (sws2b, sws2a) and double cones (rh2aβ, lws). Because rh2aα is not expressed in Midas cichlids [4,11], it was excluded from this study. It should be noted that only one individual was analysed for each time point in the exchange treatments, which precludes statistical analyses of the observed changes in cone opsin expression.

Based on qPCR data, a subset of retinas was selected for FISH to reflect the time points that were characterized by strong changes in opsin expression (blue-to-red: days 1, 9, 10, 11, 12 and 14; red-to-blue: days 1, 2, 3, 4 and 14; figure 1c). Since only sws2a and sws2b showed expression changes based on qPCR (figure 1c), we performed FISH for those two opsins. For quantifying spatial patterns of opsin expression, retinas were overlaid with an $8 \times 8$ grid and for each segment, cone cells within a

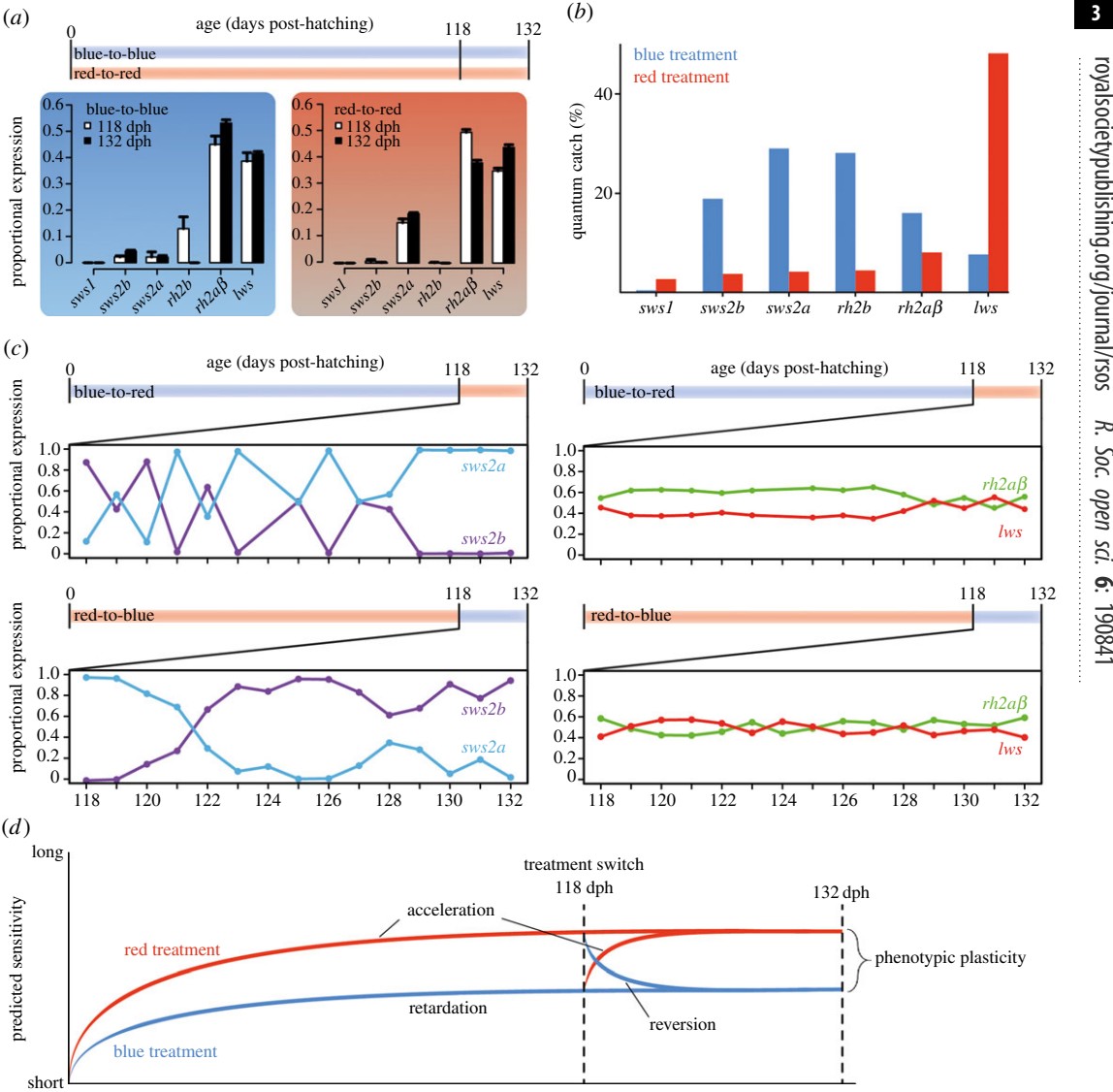

**Figure 1.** (*a*) Individuals were reared for up to 132 days, at which age they show an adult visual phenotype, under either blue or red light conditions [4]. At two time points (118 and 132 dph), proportional cone opsin expression analyses revealed cone opsin expression differences between the light treatments but also between time points. (*b*) Proportional Quantum Catch (QC) of cone opsins in the two light treatments. (*c*) At 118 dph, a subset of individuals was exchanged between the light treatments. Expression of single cone opsins changed rapidly in response to novel light condition, whereas double cone opsin expression was not affected, as measured by qPCR. (*d*) Ambient light can affect rates of ontogenetic progression in opsin expression and could even be reverted to a previous ontogenetic state (modified from [16]).

defined square (side lengths: 100 µm) were counted. We then calculated mean values for each of the eight sections along the dorsoventral axis. For further details on FISH methodology, see Karagic *et al.* [14]. Quantum catch, which represents a measure for the number of photons a visual pigment captures, was calculated for each opsin gene and light treatment based on equations from Govardovskii *et al.* [19]. Peaks of maximum light absorption ($\lambda_{max}$) were based on published Midas cichlid data (*sws2b*, *sws2a*, *rh2b*, *rh2aβ*, *lws*) [11] or from Nile Tilapia (*sws1*) [20].

## 3. Results

At an age of four months, fish reared under blue light expressed *sws2b* and *sws2a* in single cones and *rh2b*, *rh2aβ* and *lws* in double cones. By contrast, fish reared under red light expressed *sws2a* in single cones and *rh2aβ* and *lws* in double cones (figure 1*a*). When comparing cone opsin expression of fish

reared under either blue or red light for 118 and 132 days (figure 1*a*), expression of *sws2b* (Scheirer–Ray–Hare test, $p < 0.001$), *sws2a* ($p < 0.001$), *rh2b* ($p = 0.04$) and *rh2aβ* ($p = 0.049$) significantly differed between treatments whereas expression of *sws1* ($p < 0.001$), *rh2b* ($p < 0.001$) and *lws* ($p = 0.004$) differed between time points. Besides, the interaction term between treatment and age was significant for *rh2aβ* ($p = 0.001$). These results imply that ambient light can induce and maintain differential opsin expression to an age of at least four months. Since the single cone opsin *sws1* was not substantially expressed in either of the two treatments and expression of the double cone opsin *rh2b* decreased to zero in the blue treatment during the 14-day experimental period (which precluded distinguishing between normal ontogenetic changes and plasticity induced by the light treatments), those two opsins were omitted from further analyses (figure 1*a*).

Single cone opsin expression patterns (*sws2b* and *sws2a*) could be rapidly altered at an age of four months, in the same direction as the typical ontogenetic progression (blue-to-red), but could also be reverted to an earlier developmental stage (red-to-blue; figure 1*c*). In the blue-to-red treatment, expression of *sws2b* and *sws2a* was highly variable up to 10 days after the treatment switch, afterwards only *sws2a* was expressed in single cones. In the red-to-blue treatment, only *sws2a* was expressed before the treatment switch and then expression shifted to predominantly *sws2b* only 3 days after switching (figure 1*c*). Notably, expression of *rh2aβ* and *lws*, the two opsins predominantly expressed in double cones, was not affected by the drastic change in light conditions (figure 1*c*).

Fluorescent *in situ* hybridization (FISH) allowed us to obtain a more detailed picture of spatial patterning of gene expression across the retina. Whenever the single cone opsins *sws2b* and *sws2a* were expressed simultaneously, co-expression in the same photoreceptor cell was common and widespread across the whole retina (figure 2*c*). At the beginning of the treatment switch, blue-to-red fish expressed *sws2b* and *sws2a* without any obvious gradient along the dorsoventral axis and this pattern was maintained until 10 days after switching (figure 2*a*), afterwards only *sws2a* was expressed (similar to qPCR results; figure 1*c*). A different picture emerged in the red-to-blue fish where initially only *sws2a* was expressed (figure 2*b*), but already after 2 days, expression of *sws2b* was initiated mostly in the ventral part of the retina and appeared to spread dorsally afterwards. At day 4, *sws2b* expression was dominant but still lower in the most dorsal part of the retina (figure 2*b*), indicating that expression shifts were induced along a dorsoventral gradient.

# 4. Discussion

Experimental manipulations showed that raising Midas cichlids under different light conditions (blue and red) for four months caused pronounced expression differences for some cone opsins (*sws2b*, *sws2a*, *rh2b*; figure 1*a*). Moreover, at this age, Midas cichlids were still able to rapidly change single cone opsin expression when exposed to novel light conditions (figure 1*b,d*). In several cichlids, including our study species, cone opsin expression is known to shift during ontogeny from absorbing light at shorter to longer wavelengths [4,21]. In Midas cichlids, the UV-sensitive *sws1* is only expressed during the first days after hatching and is then replaced by the violet-sensitive *sws2b* and subsequently by the blue-sensitive *sws2a* [4]. This typical progression could be affected by the light conditions that individuals were reared under (figure 1*c*; [4]), by retarding or accelerating the rate of ontogenetic change (figure 1*d*). Remarkably, single cone opsin expression patterns could even be reverted to resemble a more juvenile phenotype when exposed to short-wavelength light (figure 1*d*).

Phenotypic plasticity of cone opsin expression in adult individuals has been shown only in a few teleost species, including *A. astorquii* [4,6,15]. However, it was not known whether plasticity was constrained by the ontogenetic progression from short- to long-wavelength sensitivity, as suggested by previous findings [4]. Our results demonstrate that Midas cichlids are indeed able to revert their visual phenotype to reflect a previous ontogenetic state when moved from red to blue light and shifts seemed to occur even faster than in the opposite direction (figure 1*c*). Hence, changes were not merely produced in the same direction as the ontogenetic progression but the expression of single cone opsins could be freely adjusted to different light conditions, even at an age of four months. Previously, we have shown that individuals of this species can even exhibit phenotypic plasticity of cone opsins at an age of six months [4]. At this age, *A. astorquii* commonly shows an adult visual phenotype, suggesting that cone opsin expression still exhibits phenotypic plasticity after the ontogenetic progression of the visual phenotype is complete. Crater lake Apoyo, the habitat of *A. astorquii*, is very deep [22] and water clarity changes seasonally, as in other Nicaraguan crater lakes [23]. Hence, the ability to adjust opsin expression

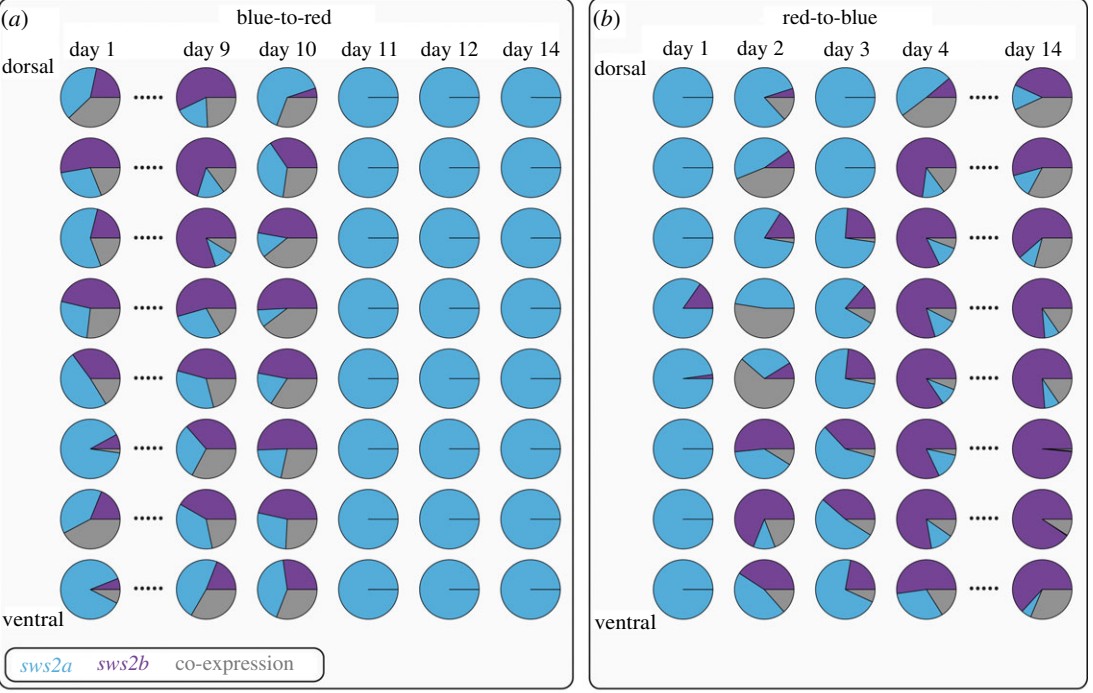

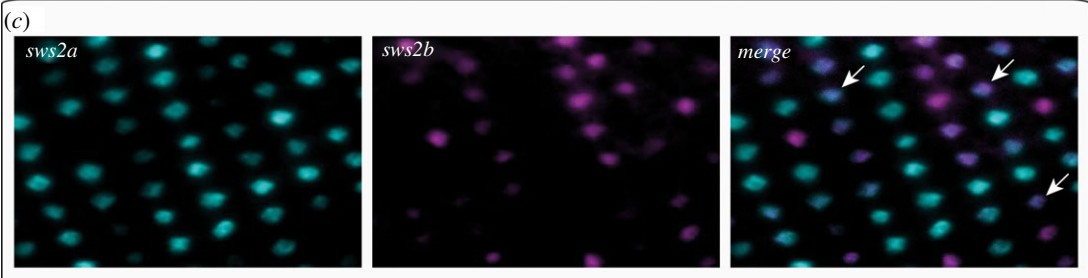

**Figure 2.** Fluorescent *in situ* hybridization (FISH) of single cone opsins (*sws2a* and *sws2b*). (*a*) In the blue-to-red treatment, *sws2a* and *sws2b* are expressed one day after the exchange, but no dorsoventral gradient in opsin expression is evident. Only after day 11, the switch to exclusively expressing *sws2a* across the whole retina is complete. (*b*) In the red-to-blue treatment, *sws2a* is almost exclusively expressed one day after the exchange but *sws2b* expression increases rapidly in the ventral part of the retina and appears to spread dorsally over time. (*c*) The transitional phase in single cone opsin expression is characterized by substantial levels of co-expression (white arrows).

appears to be important for the visual ecology of these fish and might be selectively advantageous in their natural environment.

Shifts in single cone opsin expression were initiated in the ventral retina and then spread dorsally over the course of a few days in the red-to-blue treatment. By contrast, no such gradient was evident in the blue-to-red treatment (figure 2). Dorsoventral gradients in opsin expression are widespread across vertebrates and are assumed to be adaptive as they improve visual tasks associated with different regions of the retina [24] and can also be induced when the light of varying spectral composition reaches different parts of the retina [25,26]. It would be highly interesting to further explore whether the same molecular mechanisms underlie such opsin expression gradients produced either by adaptive evolution or by phenotypic plasticity.

One question that remains is whether the observed changes in cone opsin expression are indeed adaptive. Commonly, it is assumed that cone opsin expression aims to maximize quantum catch and we have recently shown that several species of cichlid fishes indeed shifted opsin expression according to the light conditions during adaptive evolution in novel environments [11,13]. But, our models based on the light conditions in our treatments show that *sws2a* actually captures more light than *sws2b* in the blue treatment (figure 1*b*). Hence, the observed shifts in single cone expression cannot be explained by an increased quantum catch of single cone opsins alone but visual tasks such as colour differentiation or contrast detection might play a crucial role. Contrarily, the quantum catch of double cone opsins (*rh2aβ* and *lws*) strongly differs between treatments, but gene expression was

not affected (figure 1). The observation that single cone opsin expression remained highly variable across individuals until day 10 in the blue-to-red treatment (figure 1c) could, thus, be due to the fact that quantum catch of both *sws2* paralogs is low, but equal in the red light (figure 1b). This might inhibit a more rapid alteration of single cone opsin expression as seen in the red-to-blue treatment. However, at this point, it is not clear how the observed changes affect the perception of different colours and behavioural data will be needed to clarify the advantage of varying visual phenotypes in different light environments and the maintained capacity to alter them throughout ontogeny.

The rapid changes in opsin expression seen here cannot be explained by apoptosis and replacement of photoreceptors nor by the addition of novel photoreceptors due to overall eye growth since these mechanisms would take a longer time. Rather, a switch in expressed single cone opsins must have occurred within the existing photoreceptors, as was previously described during the development of salmon [27]. The mechanism presented by Cheng & Flamarique implies a transitional phase with co-expression of two opsins in the same photoreceptor. Indeed, we observed co-expression of *sws2a* and *sws2b* during our exchange experiment (figure 2), providing strong evidence that rapid adjustment of *A. astorquii's* visual system to novel light conditions occurred via replacement of the expressed opsins within single cone cells.

We provide experimental evidence that cichlid fish from the same brood maintain distinct cone opsin expression patterns until an age of at least four months when reared under different light conditions. Further, at that age, individuals can rapidly adjust colour vision by changing expression levels of single cone opsin genes when switched between these light treatments. This switch occurs via a dorsoventral gradient in one case and is characterized by a transitional phase of co-expression. Most interestingly, we found that visual sensitivity tuning can occur both towards shorter and longer wavelengths, thereby even reverting previous ontogenetic changes. This ability probably represents an adaptation to the temporally and spatially changing light conditions in the native habitat of these fish, a single crater lake in Nicaragua, by allowing them to freely and extremely rapidly adjust their visual system.

Ethics. All experiments were performed at the animal research facility of the University of Konstanz, Germany. The experiment was approved by German authorities (Regierungspräsidium Freiburg, Aktenzeichen 35-9185.81/G-16/07).
Data accessibility. All relevant data related to this article are available from the Dryad Digital Repository [28]. For publication: https://doi.org/10.5061/dryad.3b65k44.
Authors' contributions. All authors conceived the experimental design, N.K. reared fish, dissected individuals and performed FISH. A.H. performed qPCRs and analysed data together with N.K. and J.T.D. A.H. wrote the final version of the manuscript with comments from all authors.
Competing interests. The authors declare no competing interests.
Funding. This work was supported by the European Research Council through ERC-advanced (grant number 293700-GenAdap to A.M.), the Deutsche Forschungsgemeinschaft (grant number to 914/2-1 to J.T.D.) and the Young Scholar Fund of the University of Konstanz (grant number 83944318 to J.T.D.).
Acknowledgements. We thank G. Hamann for help with data collection and S. Rometsch for valuable comments on the manuscript.

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
