## [Reviewer comments · Royal Society Open Science]

Review History

RSOS-190841.R0 (Original submission)

Review form: Reviewer 1

Is the manuscript scientifically sound in its present form?

Yes

Are the interpretations and conclusions justified by the results?

Yes

Is the language acceptable?

Yes

Is it clear how to access all supporting data?

Yes

Do you have any ethical concerns with this paper?

No

Have you any concerns about statistical analyses in this paper?

Yes

Recommendation?

Accept with minor revision (please list in comments)

Comments to the Author(s)

Härer et al., conducted a study to examine the variation in opsin gene expression of the Midas cichlid as a result of ontogeny and plasticity. The authors achieved this by conducting temporal sampling and a manipulative experiment paired with qPCR and FISH analyses of opsin expression. In general, I found the paper to be clearly written. The results of the study are interesting, and I think useful (particularly visual ecologists). However, I think that a simple statistical analysis comparing the magnitude of change in opsin expression across time and between the treatments would make the interpretation of the data a little easier and the results more meaningful (outline in more detail below). Otherwise I have only a few generally minor comments.

Primary comments:

Line 43 – It would be useful to state here or elsewhere in the introduction what the opsin repertoire is for this species in terms of opsin types (SWS1, SWS2 etc).

If the authors want to suggest that the plasticity is adaptive (as suggested in the title) I think there needs to be some bolstering of this argument in the discussion. It is debatable whether or not ‘matching’ of expression to ambient conditions (to maximize quantum catch) is a signature of increased visual function as this may decrease the ability to detect contrast (which to the author’s credit is briefly mentioned in the discussion). Given that we don’t know functionally what would be more useful to these fish, quantum catch or contrast detection performance-wise, I would argue it is unclear whether this plasticity is adaptive or not. I recommend either changing the title or providing more clear evidence for the adaptive utility of the observed plasticity.

I understand that at some sampling time points statistical analysis was not possible due to the sampling of a single individual. However, there is often individual level variation in opsin expression (and sex specific variation) so it would be worthwhile to indicate the caveats of the sampling structure of the paper. Additionally, individual or sex related variation might help explain the back and forth fluctuations in expression seen for SWS2a and SWS2b in the blue-to-red treatment.

In contrast, at the beginning and end of the experimental period 6 individuals were collected and statistical tests should be possible. Were the changes in expression statistically significant? It would be worthwhile to indicate the magnitude of the changes in expression over time and between the experimental treatments (with the associated statistics) in the text. Then it would be possible to compare the relative magnitude of the shift in expression due to ontogeny vs plasticity - a more meaningful and useful metric.

The experiment looked at fish up to 4 months of age, when do these fish sexually mature? Are these ‘adult’ fish? It would be useful to state this somewhere in the paper. As perhaps this would affect the likelihood that there is a later point where opsin expression would become more permanently fixed (i.e. could no longer change).

Minor comments:

Line 23 – “respond phenotypically plastic” would make more sense as “exhibit phenotypic plasticity”

Lines 29 – 32 – The message here is a little confusing, are there two separate environments during this time? Perhaps split into two sentences.

Review form: Reviewer 2

Is the manuscript scientifically sound in its present form?

Yes

Are the interpretations and conclusions justified by the results?

Yes

Is the language acceptable?

Yes

Is it clear how to access all supporting data?

Yes

Do you have any ethical concerns with this paper?

No

Have you any concerns about statistical analyses in this paper?

No

Recommendation?

Accept with minor revision (please list in comments)

Comments to the Author(s)

The manuscript “Reverting ontogeny: rapid adaptive plasticity of color vision in cichlid fish” presented by Harer et al examines opsin gene expression in different light environments. I have been watching with great interest as the authors have been building this system over the last several years since I feel it provides a promising model for studies at the interface of evolution and visual ecology. This study presents some very interesting and novel data (particularly the FISH data coupled with the qPCR). I have only relatively minor comments that I feel the authors could easily address before the manuscript is ready for publication.

L. 24, 55, 129. It’s unclear what the author’s are trying to say – that it is unknown if all species are constrained to progression from short to long wavelength, or just *A. astoriquii*. Since the author’s cite Nandamuri et al 2017 at L.129 (in which they show another cichlid species switching to short wavelengths as adults) I’m guessing the authors meant *A. astoriquii* specifically. But this should be clarified as the authors seem to be overstepping the novelty of their findings, particularly in line 174 when they claim “This unusual ability...” which is neither unique, nor tested in many taxa. I suggest the authors tone down the ‘uniqueness’ of a switch to shorter wavelengths throughout and highlight that adult plasticity in visual tuning remains to be addressed across many taxa.

Fig1C – the y axis here is labeled as proportional expression- but it’s unclear how you went from relative to housekeeping genes to proportional single cone opsin (which I assume is what it is).

Throughout the manuscript the authors stress that they are observing plasticity after reaching an adult phenotype. However, as authors say on L 95 “expression of the double cone opsin rh2b

decreased to zero in both treatments during the 14-day experimental period (which precluded distinguishing between normal ontogenetic changes and plasticity induced by the light treatments)". This suggests their study was not actually starting from fully developed adult phenotypes. I don't think this changes the overall message of the paper, but would appreciate if the authors could acknowledge this in their discussion as it currently has contradictory data and statements.

When I went to find how expression values were calculated from the housekeeping values (authors point to their previous paper) – I was slightly alarmed to see that they have not taken into account the relative efficiencies of their different assays as this can dramatically skew their results. While I don't think it will alter the qualitative patterns that are the focus of this paper (the switch), I would strongly encourage this group to incorporate the relative assay efficiencies (for an example of how to do this properly see the paper they cited as #15 in the manuscript).

Supplementary qPCR data – In the supplementary 'qpcr_data' file there is a tab titled "single_cones_exchange" with Ct values for sws2b, sws2a, gapdh2 and imp2. But under the tab titled "double_cones_exchange" there are also Ct values for sws2b, sws2a, gapdh2 and imp2 (and values differ between tabs). I assume the "double_cones_exchange" is actually showing rh2aB and lws? Please also provide the assay efficiencies for the assays.

I am also confused as to why the values for gapdh2 and imp2 differ in the single cone and double cone data sets when they came from the same individual? Perhaps these were run on different plates?

I am a bit alarmed to see only one value for each qPCR assay. qPCR is a highly variable technique- each assay should be run in at least triplicate and variation in these reads must be reported. See Bustin et al 2009. "The MIQE Guidelines: Minimum Information for Publication of Quantitative Real-Time PCR Experiments". It is my understanding that these standards are required by the Royal Society (<https://blogs.royalsociety.org/publishing/transparent-reproducible-research/>). It is already outside the norm to use qPCR measures from just one biological replicate per time point – however I realize the patterns observed holds across days sampled and they were coupled with FISH data that backs up the qualitative qPCR findings.

Decision letter (RSOS-190841.R0)

19-Jun-2019

Dear Dr Härer

On behalf of the Editors, I am pleased to inform you that your Manuscript RSOS-190841 entitled "Reverting ontogeny: rapid adaptive plasticity of color vision in cichlid fish" has been accepted for publication in Royal Society Open Science subject to minor revision in accordance with the referee suggestions. Please find the referees' comments at the end of this email.

The reviewers and handling editors have recommended publication, but also suggest some minor revisions to your manuscript. Therefore, I invite you to respond to the comments and revise your manuscript.

- Ethics statement

If your study uses humans or animals please include details of the ethical approval received, including the name of the committee that granted approval. For human studies please also detail

whether informed consent was obtained. For field studies on animals please include details of all permissions, licences and/or approvals granted to carry out the fieldwork.

- Data accessibility

If you wish to submit your supporting data or code to Dryad (<http://datadryad.org/>), or modify your current submission to dryad, please use the following link:
<http://datadryad.org/submit?journalID=RSOS&manu=RSOS-190841>

- Competing interests

- Authors' contributions

- Acknowledgements

- Funding statement

Because the schedule for publication is very tight, it is a condition of publication that you submit

the revised version of your manuscript before 28-Jun-2019. Please note that the revision deadline will expire at 00.00am on this date. If you do not think you will be able to meet this date please let me know immediately.

If your manuscript is newly submitted and subsequently accepted for publication, you will be

asked to pay the article processing charge, unless you request a waiver and this is approved by Royal Society Publishing. You can find out more about the charges at <http://rsos.royalsocietypublishing.org/page/charges>. Should you have any queries, please contact openscience@royalsociety.org.

on behalf of Dr Michael Tobler (Associate Editor) and Kevin Padian (Subject Editor)
openscience@royalsociety.org

Associate Editor Comments to Author (Dr Michael Tobler):

We have received the feedback of two reviewers. They agree that the study is sound and suitable for publication in RSOS. I agree with reviewer 1 that suggest the addition of an additional statistical analysis. I encourage the authors to address the constructive feedback by both reviewers and a revised manuscript should be acceptable for publication.

Reviewer comments to Author:

Reviewer: 1

Comments to the Author(s)

Härer et al., conducted a study to examine the variation in opsin gene expression of the Midas cichlid as a result of ontogeny and plasticity. The authors achieved this by conducting temporal sampling and a manipulative experiment paired with qPCR and FISH analyses of opsin expression. In general, I found the paper to be clearly written. The results of the study are interesting, and I think useful (particularly visual ecologists). However, I think that a simple statistical analysis comparing the magnitude of change in opsin expression across time and between the treatments would make the interpretation of the data a little easier and the results more meaningful (outline in more detail below). Otherwise I have only a few generally minor comments.

Primary comments:

Line 43 – It would be useful to state here or elsewhere in the introduction what the opsin repertoire is for this species in terms of opsin types (SWS1, SWS2 etc).

If the authors want to suggest that the plasticity is adaptive (as suggested in the title) I think there needs to be some bolstering of this argument in the discussion. It is debatable whether or not ‘matching’ of expression to ambient conditions (to maximize quantum catch) is a signature of increased visual function as this may decrease the ability to detect contrast (which to the author’s credit is briefly mentioned in the discussion). Given that we don’t know functionally what would be more useful to these fish, quantum catch or contrast detection performance-wise, I would

argue it is unclear whether this plasticity is adaptive or not. I recommend either changing the title or providing more clear evidence for the adaptive utility of the observed plasticity.

I understand that at some sampling time points statistical analysis was not possible due to the sampling of a single individual. However, there is often individual level variation in opsin expression (and sex specific variation) so it would be worthwhile to indicate the caveats of the sampling structure of the paper. Additionally, individual or sex related variation might help explain the back and forth fluctuations in expression seen for SWS2a and SWS2b in the blue-to-red treatment.

In contrast, at the beginning and end of the experimental period 6 individuals were collected and statistical tests should be possible. Were the changes in expression statistically significant? It would be worthwhile to indicate the magnitude of the changes in expression over time and between the experimental treatments (with the associated statistics) in the text. Then it would be possible to compare the relative magnitude of the shift in expression due to ontogeny vs plasticity - a more meaningful and useful metric.

The experiment looked at fish up to 4 months of age, when do these fish sexually mature? Are these 'adult' fish? It would be useful to state this somewhere in the paper. As perhaps this would affect the likelihood that there is a later point where opsin expression would become more permanently fixed (i.e. could no longer change).

Minor comments:

Line 23 - "respond phenotypically plastic" would make more sense as "exhibit phenotypic plasticity"

Lines 29 - 32 - The message here is a little confusing, are there two separate environments during this time? Perhaps split into two sentences.

Reviewer: 2

Comments to the Author(s)

The manuscript "Reverting ontogeny: rapid adaptive plasticity of color vision in cichlid fish" presented by Harer et al examines opsin gene expression in different light environments. I have been watching with great interest as the authors have been building this system over the last several years since I feel it provides a promising model for studies at the interface of evolution and visual ecology. This study presents some very interesting and novel data (particularly the FISH data coupled with the qPCR). I have only relatively minor comments that I feel the authors could easily address before the manuscript is ready for publication.

L. 24, 55, 129. It's unclear what the author's are trying to say - that it is unknown if all species are constrained to progression from short to long wavelength, or just *A. astoriquii*. Since the author's cite Nandamuri et al 2017 at L.129 (in which they show another cichlid species switching to short wavelengths as adults) I'm guessing the authors meant *A. astoriquii* specifically. But this should be clarified as the authors seem to be overstepping the novelty of their findings, particularly in line 174 when they claim "This unusual ability..." which is neither unique, nor tested in many taxa. I suggest the authors tone down the 'uniqueness' of a switch to shorter wavelengths throughout and highlight that adult plasticity in visual tuning remains to be addressed across many taxa.

Fig1C - the y axis here is labeled as proportional expression- but it's unclear how you went from relative to housekeeping genes to proportional single cone opsin (which I assume is what it is).

Throughout the manuscript the authors stress that they are observing plasticity after reaching an

adult phenotype. However, as authors say on L 95 “expression of the double cone opsin rh2b decreased to zero in both treatments during the 14-day experimental period (which precluded distinguishing between normal ontogenetic changes and plasticity induced by the light treatments)”. This suggests their study was not actually starting from fully developed adult phenotypes. I don’t think this changes the overall message of the paper, but would appreciate if the authors could acknowledge this in their discussion as it currently has contradictory data and statements.

When I went to find how expression values were calculated from the housekeeping values (authors point to their previous paper) – I was slightly alarmed to see that they have not taken into account the relative efficiencies of their different assays as this can dramatically skew their results. While I don’t think it will alter the qualitative patterns that are the focus of this paper (the switch), I would strongly encourage this group to incorporate the relative assay efficiencies (for an example of how to do this properly see the paper they cited as #15 in the manuscript).

Supplementary qPCR data – In the supplementary ‘qpcr_data’ file there is a tab titled “single_cones_exchange” with Ct values for sws2b, sws2a, gapdh2 and imp2. But under the tab titled “double_cones_exchange” there are also Ct values for sws2b, sws2a, gapdh2 and imp2 (and values differ between tabs). I assume the “double_cones_exchange” is actually showing rh2aB and lws? Please also provide the assay efficiencies for the assays.

I am also confused as to why the values for gapdh2 and imp2 differ in the single cone and double cone data sets when they came from the same individual? Perhaps these were run on different plates?

I am a bit alarmed to see only one value for each qPCR assay. qPCR is a highly variable technique- each assay should be run in at least triplicate and variation in these reads must be reported. See Bustin et al 2009. “The MIQE Guidelines: Minimum Information for Publication of Quantitative Real-Time PCR Experiments”. It is my understanding that these standards are required by the Royal Society (<https://blogs.royalsociety.org/publishing/transparent-reproducible-research/>). It is already outside the norm to use qPCR measures from just one biological replicate per time point – however I realize the patterns observed holds across days sampled and they were coupled with FISH data that backs up the qualitative qPCR findings.

Author's Response to Decision Letter for (RSOS-190841.R0)

See Appendix A.

Decision letter (RSOS-190841.R1)

08-Jul-2019

Dear Dr Härer,

I am pleased to inform you that your manuscript entitled "Reverting ontogeny: rapid phenotypic plasticity of color vision in cichlid fish" is now accepted for publication in Royal Society Open Science.

on behalf of Dr Michael Tobler (Associate Editor) and Kevin Padian (Subject Editor)
openscience@royalsociety.org

Follow Royal Society Publishing on Twitter: [@RSocPublishing](https://twitter.com/RSocPublishing)
Follow Royal Society Publishing on Facebook:
<https://www.facebook.com/RoyalSocietyPublishing.FanPage/>
Read Royal Society Publishing's blog: <https://blogs.royalsociety.org/publishing/>

Appendix A

(RSOS-190841): Reverting ontogeny: rapid phenotypic plasticity of color vision in cichlid fish

Associate Editor Comments to Author (Dr Michael Tobler):

We have received the feedback of two reviewers. They agree that the study is sound and suitable for publication in RSOS. I agree with reviewer 1 that suggest the addition of an additional statistical analysis. I encourage the authors to address the constructive feedback by both reviewers and a revised manuscript should be acceptable for publication.

Dear Prof. Tobler, we appreciate the feedback you and the two anonymous reviewers have provided. We are responding to each issue in detail below. Please note that all the indicated line numbers refer to the main document showing the tracked changes.

Reviewer comments to Author:

Reviewer: 1

Comments to the Author(s)

Härer et al., conducted a study to examine the variation in opsin gene expression of the Midas cichlid as a result of ontogeny and plasticity. The authors achieved this by conducting temporal sampling and a manipulative experiment paired with qPCR and FISH analyses of opsin expression. In general, I found the paper to be clearly written. The results of the study are interesting, and I think useful (particularly visual ecologists). However, I think that a simple statistical analysis comparing the magnitude of change in opsin expression across time and between the treatments would make the interpretation of the data a little easier and the results more meaningful (outline in more detail below). Otherwise I have only a few generally minor comments.

We appreciate the constructive criticism.

Primary comments:

Line 43 – It would be useful to state here or elsewhere in the introduction what the opsin repertoire is for this species in terms of opsin types (SWS1, SWS2 etc).

We agree with the reviewer and incorporated information on the opsin repertoire in the introduction (lines 45-47).

If the authors want to suggest that the plasticity is adaptive (as suggested in the title) I think there needs to be some bolstering of this argument in the discussion. It is debatable whether or not 'matching' of expression to ambient conditions (to maximize quantum catch) is a signature of increased visual function as this may decrease the ability to detect contrast (which to the author's credit is briefly mentioned in the discussion). Given that we don't know functionally what would be more useful to these fish, quantum catch or contrast detection performance-wise, I would argue it is unclear whether this plasticity is adaptive or not. I recommend either changing the title or providing more clear evidence for the adaptive utility of the observed plasticity.

We agree with the reviewer that, at this point, we do not have conclusive behavioral data that clearly shows that the observed changes are adaptive. Hence, we changed the title and also rephrased parts of the manuscript to be more careful about the adaptive nature of the observed changes.

I understand that at some sampling time points statistical analysis was not possible due to the sampling of a single individual. However, there is often individual level variation in opsin expression (and sex specific variation) so it would be worthwhile to indicate the caveats of the sampling structure of the paper. Additionally, individual or sex related variation might help explain the back and forth fluctuations in expression seen for SWS2a and SWS2b in the blue-to-red treatment.

We now indicate in the Methods section that our sampling design of one individual per time-point precludes statistical analyses of opsin expression during the experimental period (lines 86-88). We agree that there can be quite some variation among individuals (including sex-specific differences) which might explain the high variation in the blue-to-red treatment. However, since we don't see such fluctuations in the red-to-blue treatment, we conclude that these fluctuations are most likely due to random variation among individuals but might be rather explained by differences in quantum catch as we argue in the discussion (lines 172-175).

In contrast, at the beginning and end of the experimental period 6 individuals were collected and statistical tests should be possible. Were the changes in expression statistically significant? It would be worthwhile to indicate the magnitude of the changes in expression over time and between the experimental treatments (with the associated statistics) in the text. Then it would be possible to compare the relative magnitude of the shift in expression due to ontogeny vs plasticity - a more meaningful and useful metric.

The reviewer is right that we have sampled six individuals at the beginning and the end of the experimental period. While the main focus of the paper is the rapid expression changes of certain cone opsin after switching the light conditions, we only have six individuals from groups that were constantly maintained either in blue or red light but not from any of the exchange treatments. However, we agree that it might be interesting to incorporate statistics to see how expression of certain opsins is affected long-term by raising fish under different light conditions until an age of four months and also how it might change during our experimental period. Hence, we incorporated Scheirer-Ray-Hare tests (non-parametric alternative to ANOVA) to test the effects of light treatment and time-points on expression of each cone opsin (lines 107-111). As mentioned above, we don't think that our conclusions are dependent on these results, thus, we only mention it briefly in the first paragraph of the Discussion section (lines 139-140) but don't discuss it further to maintain the manuscript short.

The experiment looked at fish up to 4 months of age, when do these fish sexually mature? Are these 'adult' fish? It would be useful to state this somewhere in the paper. As perhaps this would affect the likelihood that there is a later point where opsin expression would become more permanently fixed (i.e. could no longer change).

*We incorporated the information on the age at which these fish commonly show an adult visual phenotype in the revised version of the manuscript (lines 156-159). We agree with the reviewer that phenotypic plasticity might be reduced at an older age, however, previous work by our research group has shown that *A. astorquii* can exhibit phenotypic plasticity at an age of 6 months when the visual phenotype of a sexually mature individual is reached (Härer et al. 2017 Molecular Ecology).*

Minor comments:

Line 23 – “respond phenotypically plastic” would make more sense as “exhibit phenotypic plasticity”

Lines 29 – 32 – The message here is a little confusing, are there two separate environments during this time? Perhaps split into two sentences.

We changed these minor points according to the reviewer's suggestion.

Reviewer: 2

Comments to the Author(s)

The manuscript "Reverting ontogeny: rapid adaptive plasticity of color vision in cichlid fish" presented by Harer et al examines opsin gene expression in different light environments. I have been watching with great interest as the authors have been building this system over the last several years since I feel it provides a promising model for studies at the interface of evolution and visual ecology. This study presents some very interesting and novel data (particularly the FISH data coupled with the qPCR). I have only relatively minor comments that I feel the authors could easily address before the manuscript is ready for publication.

L. 24, 55, 129. It's unclear what the author's are trying to say – that it is unknown if all species are constrained to progression from short to long wavelength, or just *A. astoriquii*. Since the author's cite Nandamuri et al 2017 at L.129 (in which they show another cichlid species switching to short wavelengths as adults) I'm guessing the authors meant *A. astoriquii* specifically. But this should be clarified as the authors seem to be overstepping the novelty of their findings, particularly in line 174 when they claim "This unusual ability..." which is neither unique, nor tested in many taxa. I suggest the authors tone down the 'uniqueness' of a switch to shorter wavelengths throughout and highlight that adult plasticity in visual tuning remains to be addressed across many taxa.

*We agree that phenotypic plasticity in opsin expression has been described before but has not been found in many species, but, it might not be that unusual once it gets tested in more species so we removed the word "unusual" from the text (line 203). Moreover, we understand the reviewer's concerns but we want to argue that the point we are making about reverting ontogeny remains valid in our eyes. The point we are making is not whether adult individuals are able to change opsin expression towards shorter wavelengths per se but it is rather about whether this can happen in the opposite direction of natural ontogenetic changes. Midas cichlids, as shown by our group before (Härer et al. 2017 Molecular Ecology), but also many African cichlids (reviewed in Carleton et al. 2016 Genesis) undergo changes in opsin expression from short to long wavelength sensitivity. However, the study species used by Nandamuri are not one of those species as they show a short wavelength sensitive visual phenotype throughout their life and don't change cone opsin expression during development. So yes, the *Metriaclima* species from Nandamuri et al. 2017 might be able to change opsin expression towards shorter wavelengths, however, this cannot be related to ontogeny. Hence, we would like to emphasize that we still think that our findings of phenotypic plasticity reverting ontogenetic changes in opsin expression are novel and have not been shown, to the best of our knowledge, in any other species.*

Fig1C – the y axis here is labeled as proportional expression- but it's unclear how you went from relative to housekeeping genes to proportional single cone opsin (which I assume is what it is).

The legend in Fig. 1C stating "proportional expression" is actually correct. However, we realized that the corresponding part of the Methods section was outdated as we changed the analyses to calculate proportional expression of the two single cone (sws2b & sws2a) and double cone opsins (rh2aβ & lws) rather than normalizing with housekeeping genes. This type of analysis gives a much better idea and visualization of which opsin is mainly expressed and how the proportions change

over time. Hence, we changed the corresponding part in the Methods section but also in the Supplementary Material.

Throughout the manuscript the authors stress that they are observing plasticity after reaching an adult phenotype. However, as authors say on L 95 “expression of the double cone opsin rh2b decreased to zero in both treatments during the 14-day experimental period (which precluded distinguishing between normal ontogenetic changes and plasticity induced by the light treatments)”. This suggests their study was not actually starting from fully developed adult phenotypes. I don’t think this changes the overall message of the paper, but would appreciate if the authors could acknowledge this in their discussion as it currently has contradictory data and statements.

We understand the reviewer’s concern that talking about an adult visual phenotype might have created confusion. However, we are convinced that our argument of observing plasticity after reaching an adult phenotype still holds, particularly when looking at the red to blue treatment. At both ages of 118 & 132 dph, the fish raised under red light expressed sws2a, rh2a β and lws, which constitutes the adult phenotype in Midas cichlids. However, when putting those fishes in blue light, they reverted to a previous state as we claim in the manuscript (lines 139-141). However, we acknowledge that the fish raised under blue light seem to have a delayed ontogenetic progression and still change in opsin expression between 118 & 132 dph. But we want to emphasize that we do not claim that these fishes reached an adult visual phenotype anywhere in the manuscript but we now made it more clear that expression of rh2b only changed in the blue treatment (lines 113-114).

When I went to find how expression values were calculated from the housekeeping values (authors point to their previous paper) – I was slightly alarmed to see that they have not taken into account the relative efficiencies of their different assays as this can dramatically skew their results. While I don’t think it will alter the qualitative patterns that are the focus of this paper (the switch), I would strongly encourage this group to incorporate the relative assay efficiencies (for an example of how to do this properly see the paper they cited as #15 in the manuscript).

As mentioned further above, we actually did not use housekeeping genes for normalization but rather only calculated proportional expression of cone opsins genes. Regarding the point of taking into account qPCR efficiencies for the different genes, we have calculated efficiencies for all genes and the efficiencies vary between 95 & 103 % showing that there are no strong differences in efficiency that might substantially bias our results. However, we reanalyzed our data incorporating efficiencies in the new version of the manuscript and also updated all qPCR results shown in Figure 1 in this context. The efficiencies are noted in the Supplementary Material.

Supplementary qPCR data – In the supplementary ‘qpcr_data’ file there is a tab titled “single_cones_exchange” with Ct values for sws2b, sws2a, gapdh2 and imp2. But under the tab titled “double_cones_exchange” there are also Ct values for sws2b, sws2a, gapdh2 and imp2 (and values differ between tabs). I assume the “double_cones_exchange” is actually showing rh2aB and lws? Please also provide the assay efficiencies for the assays.

We corrected the labelling of the columns in the supplementary file and as explained further above, we have removed the housekeeping gene columns as we did not take them into account for the final analysis of our data.

I am also confused as to why the values for gapdh2 and imp2 differ in the single cone and double cone data sets when they came from the same individual? Perhaps these were run on different plates?

This can be explained by the fact that we used different cDNA dilutions for measuring single and double cone expression but, as mentioned above, we did not use the housekeeping genes in the end.

I am a bit alarmed to see only one value for each qPCR assay. qPCR is a highly variable technique- each assay should be run in at least triplicate and variation in these reads must be reported. See Bustin et al 2009. "The MIQE Guidelines: Minimum Information for Publication of Quantitative Real-Time PCR Experiments". It is my understanding that these standards are required by the Royal Society (<https://blogs.royalsociety.org/publishing/transparent-reproducible-research/>). It is already outside the norm to use qPCR measures from just one biological replicate per time point – however I realize the patterns observed holds across days sampled and they were coupled with FISH data that backs up the qualitative qPCR findings.

In the last version of our manuscript, we were not completely clear about the way we collected the qPCR data. Actually, we did use two (or three in few cases) technical replicates and only reported the mean values in the Supplementary Data. We have now altered the Supplementary Data and incorporated all raw Ct values of the technical replicates.